# Neuroprotective Strategies and Cell-Based Biomarkers for Manganese-Induced Toxicity in Human Neuroblastoma (SH-SY5Y) Cells

**DOI:** 10.3390/biom14060647

**Published:** 2024-05-31

**Authors:** Catherine M. Cahill, Sanjan S. Sarang, Rachit Bakshi, Ning Xia, Debomoy K. Lahiri, Jack T. Rogers

**Affiliations:** 1Neurochemistry Laboratory, Massachusetts General Hospital, Harvard Medical School, Boston, MA 02129, USA; ccahill@mgh.harvard.edu (C.M.C.); sanjansarang1@gmail.com (S.S.S.); rbakshi1@mgh.harvard.edu (R.B.); nxia@mgh.harvard.edu (N.X.); 2Department of Psychiatry and Medical & Molecular Genetics, Indiana Alzheimer’s Disease Research Center, Stark Neurosciences Research Institute, Indiana University School of Medicine, Indianapolis, IN 46202, USA; dlahiri@iupui.edu

**Keywords:** manganese neurotoxicity, neuroprotection, urate, inflammation, oxidative stress, untranslated regions, mRNAs, amyloid precursor protein (APP), ferritin, Parkinsonism, metallothionein-III

## Abstract

Manganese (Mn) is an essential heavy metal in the human body, while excess *Mn* leads to neurotoxicity, as observed in this study, where 100 µM of *Mn* was administered to the human neuroblastoma (SH-SY5Y) cell model of dopaminergic neurons in neurodegenerative diseases. We quantitated pathway and gene changes in homeostatic cell-based adaptations to *Mn* exposure. Utilizing the Gene Expression Omnibus, we accessed the GSE70845 dataset as a microarray of SH-SY5Y cells published by Gandhi et al. (2018) and applied statistical significance cutoffs at *p* < 0.05. We report 74 pathway and 10 gene changes with statistical significance. ReactomeGSA analyses demonstrated upregulation of histones (5 out of 10 induced genes) and histone deacetylases as a neuroprotective response to remodel/mitigate *Mn*-induced DNA/chromatin damage. Neurodegenerative-associated pathway changes occurred. NF-κB signaled protective responses via Sirtuin-1 to reduce neuroinflammation. Critically, *Mn* activated three pathways implicating deficits in purine metabolism. Therefore, we validated that urate, a purine and antioxidant, mitigated *Mn*-losses of viability in SH-SY5Y cells. We discuss *Mn* as a hypoxia mimetic and trans-activator of HIF-1α, the central trans-activator of vascular hypoxic mitochondrial dysfunction. *Mn* induced a 3-fold increase in mRNA levels for antioxidant metallothionein-III, which was induced 100-fold by hypoxia mimetics deferoxamine and zinc.

## 1. Introduction

The central nervous system is subject to acute, chronic, and latent perturbation by heavy metals, including mercury (*Hg*), lead (*Pb*), and manganese (*Mn*) [1]. The acute neurotoxicity of *Hg* and *Pb* is well known. Both of these metals are also implicated in Alzheimer’s disease (AD) [2,3,4]. *Mn* is associated with several neurodegenerative conditions, including movement disorders such as Parkinson’s disease (PD) [5,6]. There are multiple pleiotropic mechanisms of manganese toxicity within the nervous system [7,8]. At the cellular level, we previously reported pathways by which this divalent cationic metal disrupts translation of the L- and H-subunits of the iron storage multimer ferritin and the Alzheimer’s amyloid β (Aβ) precursor protein (APP) that activates iron efflux by ferroportin (FPN) [7,9,10,11,12]. The contribution of APP to iron homeostasis as a ferroportin-binding protein remains to be solved [9,13]. Nevertheless, perturbations of iron-dependent post-transcriptional regulatory events to both ferritin and APP are associated with damage to neurons by ferroptosis and AD [14,15,16]. *Mn* inhibited expression of APP at the level of translation via Iron-Responsive Elements (IREs) in 5′untranslated region (5′-UTR) sequences in its transcript, while IREs in other transcripts, such as transferrin receptor, are also potentially important regulatory determinants of AD [7,17]. The APP 5′-UTR IRE is part of a multi-target nexus that also responds to interleukin-1 and microRNA-346 [18]. This action of *Mn* can be modeled to embargo of iron export sufficient to cause ferroptosis by limiting the export of excess *Mn* and *Fe* by FPN [9,16]. Nevertheless, reduced levels of APP would also be predicted to limit Aβ accumulation after cleavage from the precursor, although monomer concentration is set by secretase cleavage and clearance [19,20,21]. Finally, the intracellular losses of both APP and ferritin significantly reduced the viability of neurons during *Mn* exposure [7]. 

Mn is an essential trace element at physiological concentrations, acting as a cofactor in many enzymatic reactions in humans, including the antioxidant enzyme superoxide dismutase [22]. Nevertheless, at elevated industrial concentrations, *Mn* damages the nervous system, causing a neurotoxic ailment that has been documented for about 150 years. Excess manganese (Mn) intake causes manganism, a disease of occupational *Mn* overexposures. Those with this chronic industrial occupational condition exhibit psychiatric and motor disturbances [23], resulting in symptoms of bradykinesia, rigidity, tremor, gait disturbance, postural instability, and dystonia and/or ataxia [24]. Elevated *Mn* in environmental water supplies also impairs childhood cognitive performances and IQ in less-developed countries that lack appropriate public water filtration capabilities [25,26], similar to better-known effects of *Pb* [27]. We reported on biomarkers of environmental manganese exposure and associations with childhood neurodevelopment [26], along with associations between APP and autism [28,29,30,31]. *Mn* excess is associated with rare cases of Parkinson’s disease (PD) in the field [23], and also results from mutations that prevent the Park-9 protein from detoxifying *Mn* in affected neurons and a-synuclein fibril formation [32,33]. 

Depending on the duration and amount of exposure, *Mn*^2+^ induces toxic reactive oxygen species in rats [34,35], while its exposure to cultured cells can facilitate toxic cellular *Fe* overload, since FPN and iron efflux are inhibited [36]. Excess toxic *Mn* exposure certainly causes motor deficits [37] as well as partial activation of GABAergic neurons in the Globus pallidus and Substantia nigra brain subregions [38]. In order to offset toxic *Mn* overload, the universal brain iron transporter FPN can function as a manganese exporter [39,40,41].

In seeking a therapy to offset *Mn* neurotoxicity, we tested the antioxidant purine urate, which exerts a currently undefined protective action towards neurons in cell-based models of Parkinson’s disease (PD) [42]. In this report, we addressed how increased neuronal urate might therapeutically oppose *Mn* toxicity, since our bioinformatic analysis demonstrated that, first, *Mn* disrupts purine metabolism via depurination and inhibition of HPRT [43]. In fact, purines are metabolically converted to uric acid by xanthine oxidase, an enzymatic activity that, in excess, causes gout. In the PD brain, however, at physiological pH, the ionized urate salts protect against 1-methyl-4-phenyl-pyridinium (MPP+) degeneration of dopaminergic neurons. In most mammals, urate is converted to allantoin by uricase (uroxidase; UOx), an enzyme primarily expressed in the liver [44,45,46,47]. Urate added to cultured neurons 24 h before and during treatment with MPP+ toxin attenuated the loss of enriched dopaminergic neurons and fully prevented their atrophy in neuron–astrocyte cultures [48]. Uric acid provides an antioxidant defense in humans that has a protective effect against oxidant and radical-caused aging and cancer [49,50]. Similar functions of uric acid and ascorbate were reported in humans [50]. UO knockout mice express brain urate at elevated levels and are less vulnerable than wild-type mice to Parkinsonian-specific neurodegenerative brain lesions [47,51]. 

In this communication, we provide a compelling rationale to further test urate’s capacity to mitigate manganese-induced liabilities to cell survival, as outlined in the bioinformatic and pilot cell-based data provided. *Mn* acts as a hypoxia mimetic in several cell types, including SH-SY5Y cells [52,53,54,55]. Therefore, in this report, we also highlighted the microarray-oriented bioinformatic findings, showing that metallothionein-III (MT-III) is a prominent *Mn*-induced transcript. We discuss MT-III may provide further cell-based antioxidant responses to mitigate *Mn*-induced ferroptosis and neurotoxicity [54,55,56,57,58].

## 2. Materials and Methods

### 2.1. Data Access

The data for the human neuroblastoma cell line (SH-SY5Y) exposed to manganese were accessed using accession GSE70845 in the Gene Expression Omnibus (GEO). This dataset was derived from SH-SY5Y cells that were chronically exposed to 100 µM (1/5 of LD_50_) of manganese as *MnCl_2_* for 24 h; the experiment was performed in triplicate. Total RNA was harvested using TRIzol, and 100 ng was used to prepare biotinylated complementary RNA (cRNA). The cRNA was hybridized on Affymetrix Prime view Human arrays and scanned using the Affymetrix GeneChip scanner, uploaded to GEO [56].

### 2.2. Data Exploration

The normalized data were downloaded from GEO and analyzed using the GEO2R bioinformatics software (https://www.ncbi.nlm.nih.gov/geo/geo2r/, accessed on 6 October 2023) [59]. The three Mn exposure samples were assigned to the manganese treatment group, and the three control samples were assigned to the control group. These groups were analyzed using linear models for microarray analysis for statistical visualization [60]. 

### 2.3. Pathway and Gene Expression Statistical Analysis

ReactomGSA [61] was used to evaluate pathway and gene expression changes following manganese exposure. The Correlation-Adjusted Mean Rank gene set analysis (CAMERA) [62] was used to conduct a differential expression analysis to determine differentially expressed pathways and genes between two sample groups. The Binomial Test was used to evaluate the statistical significance for each hit pathway and gene analysis and a Benjamini–Hochberg *p* adjustment was subsequently utilized to manage the false discovery rate. Using the filter of adjusted *p* < 0.05 as a statistical significance cut-off, the top differentially expressed genes with adjusted *p* < 0.05 were plotted onto a heatmap and hierarchically clustered using the Morpheus matrix visualization and analysis software (https://software.broadinstitute.org/morpheus/, accessed on 11 October 2023). 

### 2.4. Experimental Validation of Bioinformatic Model

Cell culture methods: The human neuroblastoma cell line SH-SY5Y was obtained from the American Type Culture Collection (ATCC) [63,64].

MTT viability assays: In six separate cotreatment assays, cells were cotreated for 72 h with 100 µM UA and separate increasing dosages of 25 µM *Mn* as *MnCl_2_*, 50 µM *Mn* as *Mn*Cl_2_, 75 µM *Mn* as *MnCl_2_*, and 100 µM *MnCl_2_*. In octuplicate pre-treatment assays, cells were pre-treated for 24 h with UA at 3 µM, 30 µM, 75 µM, 150 µM, and 300 µM. Then, the cells were washed and dosed with 100 µM *Mn* as *MnCl_2_*. After the *Mn* treatment period for both assay types, cell viability was measured by a colorimetric assay either by using the MTT (thiazolyl blue tetrazolium, Sigma-Aldrich (St. Louis, MO, USA)) viability assay or the MTS-Assay (CellTiter96 AQ assay, Promega [Madison, WI]) according to the protocol of the supplier, as reported previously [7]. In parallel, cells were grown in 12-well plates for cell growth analysis. Repetitive aliquots were taken and counted three times using a hemocytometer with trypan blue exclusion assay, as previously reported. For each treatment condition, triplicate wells were counted, and values were averaged. 

Western blotting: SH-SY5Y cells (9 × 10^6^), at a 60–70% confluency, were treated with the following conditions in duplicate, and all for 48 h. The APP Western blot had (i) Lanes 1–2: control; (ii) Lanes 3–4: 100 µM *Mn* as 100 µM *MnCl_2_*; (iii) Lanes 5–8: 2 sets of increasing UA at 75, 300 µM; and (iv) Lanes 9–12: 2 sets of 100 µM *Mn* and increasing UA at 75, 300 µM. The H-ferritin Western blot had (i) Lanes 1–2: control; (ii) Lanes 3–4: *Fe* as 100 µM ferric ammonium citrate (*FAC*); (iii) Lanes 5–6: 100 µM *Mn* as 100 µM *Mn*Cl_2_; (iv) Lanes 7–8: 100 µM UA; (v) Lanes 9–10: 100 µM *FAC* and 100 µM UA; and (vi) Lanes 11–12: 100 µM *Mn* and 100 µM UA. After subsequent washing in PBS, cells were scraped into RIPA lysis buffer. Total protein concentrations were analyzed using Bio-Rad protein assay. Cell extracts were immunoblotted as described elsewhere [7,63]. Western blots were repeated at least three times, and densitometric analysis was performed using ImageJ software (version 1.54 g) [65]. 

RT-PCR: Triplicate assays were performed in untreated versus deferoxamine (*DFO*)- and zinc-treated SH-SY5Y cells for 6 and 16 h timepoints with the following conditions (i) control; (ii) control; (iii) 100 µM *DFO*; (iv) 200 µM *Zn* as *ZnSO_4_*; (v) 100 µM *Zn* as *ZnSO_4_*; (vi) 200 µM *Zn* as *ZnSO_4_* and 100 µM *DFO*; and (vii) 100 µM *Zn* as *ZnSO_4_* 100 µM *DFO*. Total RNA was isolated using TRI-reagent (Sigma-Aldrich) according to the manufacturer’s instructions. Assays were performed using an ABI Prism 7000 system (Applied Biosystems, Foster City, CA, USA). Metallothionein-III mRNA primers, Forward: 5′AGT GCG AGG GAT GCA AAT G 3′. Reverse: 5′ACA CAC AGT CCT TGG CAC ACT T3′. APP mRNA primers (forward, 5-GCCCTGCGGAATTGACAAG-3; reverse, 5-CCATCTGCATAGTCTGTGTCTG-3). transferrin receptor mRNA primers (forward, 5-GGCTACTTGGGCTATTGTAAAGG-3; reverse, 5-CAGTTTCTCCGACAACTTTCTCT-3); and beta-actin mRNA primers (forward, 5-CATGTACGTTGCTATCCAGGC-3; reverse, 5-CTCCTTAATGTCACGCACGAT-3) were purchased from Life Technology. 

### 2.5. Experimental Statistics

Following the MTT co-treatment assays, one-tailed Student’s *t*-tests were conducted between *Mn* and *Mn* + UA treatment groups. Statistical significance was deemed to be *p* < 0.05, indicating that UA recovery of cells did not occur by chance. Pre-treatment assays were evaluated for statistical significance via the use of one-tailed Student’s *t*-tests for control versus treatments. Using densitometric data, unpaired two-tailed Student’s *t*-tests were performed for control versus treatment lanes, as indicated by the Western blot treatment conditions. 

## 3. Results

### 3.1. Data Exploration

In the dataset analyzed by GEO2R, each sample was log_2_ normalized with the median-centering values being present, indicating cross-comparability between samples in the dataset (Figure 1a). We visualized 27,770 transcripts by volcano plot and found that 29 of them were significantly differentially expressed (adjusted *p* < 0.05). These transcripts mapped to 22 differentially expressed genes (DEGs) (Figure 1b). Of the 22 DEGs, 21 were upregulated and one was downregulated following chronic *Mn* exposure. Similarly, the mean difference (MD) plot represents the differential expression showing the most differentially expressed genes on the right region of the plot (Figure 1c).

### 3.2. Manganese Pathway Modulation 

We found significant changes via the ReactomeGSA pathway analysis tool utilizing the efficient CAMERA algorithm, which further filters DEGs from GEO2R analyses. Table 1 shows six pathways to be upregulated in *Mn* exposure; these pathways are participants in histone deacetylase (HDAC) expression as a neuroprotective response to DNA damage, depurination, and amyloidosis. Two pathways were downregulated, resulting from mitochondrial dysfunction and a protective oxidative stress pathway. 

### 3.3. Manganese Gene Modulation

Table 2 represents the top gene changes following 24 h of manganese treatment of SH-SY5Y cells. After ReactomeGSA CAMERA analysis, the 22 DEGs identified in GEO2R were narrowed down to 10 DEGs; these genes were involved in and indicate neuroprotection, histone expression, mitochondrial stress response, and detrimental expression of proinflammatory cytokines.

### 3.4. DEG Similarity and Expression Analysis 

A heatmap of the DEGs (Figure 2) represents interrelationships via a dendrogram created using one minus the Pearson correlation and complete linkage. Clustering identified that genes have potential co-expression with histone-related and cellular stress-response genes. 

### 3.5. Experimental Validation

We found that multiple pathways were activated after *Mn* exposure, including sirtuin-dependent rRNA translation and inhibition of NF-κB, as well as the capacity to inhibit pro-oxidant pathways via *NRF-2* (Table 1). However, the most significant effect of manganese exposure was to perturb three major pathways involving purine metabolism (nucleotide salvage defects were highly expressed, as well as pathways involving cleavage and damage to purines and also depurination steps of DNA and RNA). Therefore, we further tested whether pre- and co-incubation of the antioxidant purine urate might mitigate manganese toxicity (Figure 3, Figure 4 and Figure 5). The top panel in Figure 3 shows MTT assay results indicating that co-treatment of cells with uric acid (100 mM UA) mitigated manganese toxicity to SH-SY5Y neuroblastoma cells relative to controls (i.e., UA reduced the toxic effects of treatments with 25 mM, 50 mM, 75 mM, and 100 mM *Mn*Cl_2_ to cells for 72 h).

The top right-hand panel in Figure 3 exhibits a representative experiment demonstrating that the presence of urate completely mitigated a 33% loss of SHY5Y cell viability over the three-day period of exposure. This representative experiment was conducted to high statistical significance (n = 6), *p* < 0.05) as indicated in the histogram on the left-hand side (Figure 3, next page).

In terms of statistical significance, we observed a 33% loss of viability at the max dose of 100 mM manganese relative to untreated controls, while the presence of UA (100 mM) mitigated 33% of the loss of viability (MTT @ 72 h *** *p* = 0.000165; Student’s *t*-test conducted between *Mn* and *Mn* + UA groups). We noted that at 72 h, cotreatment of the cells with 100 mM urate completely rescued the cells from such toxicity. Losses of cellular viability were dose-responsive, and the degree of cell death and recovery with urate co-treatment was statistically significant (*p* < 0.05, n = 6) over several separate assays. 

As we observed in a previous publication [7], *Mn* reduced the expression of neuroprotective APP and ferritin. Western blots are shown in Figure 3 and densitometric quantitation is provided in the histogram in Figure 4 (See Appendix A for full-length gels). In three separate experiments performed in duplicate, we observed that both APP and ferritin were reduced by >50% after 48 h *Mn* exposure. During these studies, we noted that the presence of urate co-treatment restored the steady-state levels of APP, although this was not the case for steady-state levels of H-ferritin during urate + *Mn* co-exposures of SH-SY5Y cells. Densitometric quantitation of APP and H-ferritin expression relative to that of b-actin is graphically represented in Figure 4. The results of treatments with *Mn* alone and *Mn* in the presence of urate are also depicted. Duplicate assays in Figure 4A showed that *Mn* reduced APP levels by 3-fold, while the presence of urate enhanced APP protein levels 2–3-fold, such that the presence of co-added urate mitigated *Mn* inhibition of APP levels to 90 percent of control levels. We quantitated that 100 mM iron as ferric ammonium citrate (*FAC*) treatment increased ferritin expression by 2.2-fold, as standardized to b-actin in the representative densitometry of the averaged duplicate lanes. Contrary to APP expression, the presence of urate did not change the *Mn* inhibition or *Fe* induction of H-ferritin (n = 3) (Figure 4B). 

In addition to urate co-treatment with *Mn* exposures (Figure 3 and Figure 4), we also tested whether pre-treatment with urate might offset *Mn* neurotoxicity (Figure 5). Indeed, 24 h pre-treatment with urate dose afforded 14% neuroprotection to SH-SY5Y cells exposed to *Mn* (100 mM 24 h) [54,66].

Our bioinformatics findings showed a three-fold increase in MT-III mRNA (Table 2) when characterizing the transcriptomic profile of manganese-exposed SH-SY5Y cells. Manganese is a divalent cation and may well exhibit biology typical of other divalent cations, whereas *Mn*, cobalt, zinc, and DFO are known hypoxia mimetics [52,53]. Wang et al. demonstrated that hypoxia is an active inducer of MT-3 mRNA via RT-PCR analyses in human adipocytes [53].

To follow up, we demonstrated RT-PCR data in Figure 6 to show that MT-3 mRNA, transferrin receptor (TfR) TfR-mRNA and APP mRNAs were all induced in response to *DFO* and *Zn* treatments (hypoxia mimetics). We obtained three independent RT-PCR measurements from SH-SY5Y cells to quantify inductions of MT-III mRNA in order to determine whether they occurred as a result of treatment of cells with the hypoxia mimetics *DFO* and *Zn*. RT-PCR data (N = 3) showed that MT-3 mRNA levels were induced by 500-fold by *DFO* > 3–30-fold inductions of MT-3 mRNA by *ZnCl_2_* (Figure 6).

We noted from the RTPCR experiments in Figure 6 that DFO increased levels of mRNA for neuroprotectant MT-III by 500-fold (N = 3). The divalent cation zinc, another hypoxia mimetic, increased MT-3 mRNA by up to 30-fold in SH-SY5Y cells. Transferrin receptor mRNA and APP mRNA are known to be abundant transcripts. In these experiments, DFO and zinc were found to increase TfR mRNAs to a less inductive, although similar, extent relative to MT-3 mRNA when responding to hypoxia mimetic (DFO and Zn). APP mRNA is regulated via translational control circuits, and thus its mRNA was unchanged by DFO and *Mn* and Zn, as predicted.

Our bioinformatic and pre- and post-published experimental data are shown in Table 3 to quantitate, list, and rank the cellular biomarkers of *Mn* toxicity. We reported a notable increase in the transferrin receptor (TfR) mRNA in reference [7] in *Mn*Cl_2_-exposed SHSY5Y cells. However, in this study, we statistically reevaluated the source q-RT-PCR data to demonstrate a precise 7.8-fold increase in TfR mRNA levels (N = 3 standardized to b-actin controls). Iron regulatory protein 1 (IRP1) was also noted to be slightly increased at the basal level with enhanced IRE binding after *Mn* exposures (pre-published data). Iron-regulatory Protein 2 (IRP2) decreased to 9.05% (St dev (SD), ±1.47%) of controls by the same *Mn* exposures [7]. Table 3 presents our reanalysis of densitometry from Western blots to confirm our previous report [7] that H-ferritin deceased to 5% of that exhibited by the controls in SHSY5Y cells treated with 100 mM *MnCl_2_*. By contrast, a study of welders exposed to airborne *Mn* showed serum ferritin, which is rich in the L-subunit and was slightly increased by chronic *Mn* exposures while also providing an existing biomarker for iron overload [67]. Metallothionein 3 (MT-3) also implicated its role as a potential manganese chelator, elevating its expression as a neuroprotectant 3.36-fold, as listed in Table 3. Alpha-synuclein protein (60 KDa tetramer) levels decreased 2-fold in *Mn*-exposed SH-SY5Y cells.

We propose that the ratio of APP protein/APP mRNA constitutes a top candidate cellular biomarker for *Mn* toxicity. In Table 3, APP mRNA increased by 4.02-fold (10 mM, 24 h) and 3.68-fold (100 mM, 24 h) in response to *Mn* treatments of SH-SY5Y cells while APP protein was greatly inhibited, i.e., by 15% (10 mM *Mn*Cl_2_, 24 h), 50% (50 mM *Mn*Cl_2_, 24 h), 85% (100 mM *Mn*Cl_2_, 24 h) in the same SH-SY5Y cell lineage. Therefore, the ratio of APP protein/APP mRNA (index ratio of APP gene expression) provides a clear cellular biomarker signal unique to *Mn* relative to iron exposure (i.e., *Mn* causes APP translational repression, while APP mRNA levels are increased by *Mn* exposures). 

## 4. Discussion

We conducted bioinformatic analyses based on microarray data in triplicate to identify the metabolic pathways modulated by *Mn* exposures in SH-SY5Y cells [56]. Here, we report which gene transcripts are indeed activated by *Mn* exposure in SH-SY5Y cells. The study pertains to inductions in human cells in SHSY5Y, while comparative findings for mouse cell lines, including mouse N2a neuroblastoma cells, will be investigated. Although we have reported our work here using SH SY5Y cells, our lab has used other neuroblastoma lines, such as SK-N-SH, derived from the same stock as well as human primary neuronal enriched cultures (18). Based on our experience, the effect we report here most likely would work in other neuroblastoma cells; however, it needs additional future work to confirm the effect in other neuro-derivative cultures.

This is particularly relevant, since *Mn* influences APP expression via the human IRE-type-II sequences in the 5′UTR of its transcript, which is relevant to this study. Our findings in Figure 3 and Figure 4 are consistent with the model that the human APP 5′UTR-specific iron-responsive element-Type-II drives APP translation as the major conduit though which urate operates, providing therapeutic value to offset Mn toxicity (Table 3). Our current findings support future tests to establish species-specific differences by which *Mn* exposures might induce APP translation as a biomarker for predicting toxicity in human SHSY5Y cells compared to mouse cell N2A types and ultimately in vivo. There may be differences for manganese toxic actions acting through the APP 5′UTR sequences in humans compared to mouse models (Appendix A). 

Our bioinformatic findings were experimentally validated via MTT viability assays and RT-PCR analyses. We found, as a first example, that *Mn* activated histone deacetylases, which act to prevent chromatin condensations from oxidative stresses induced by *Mn*. Consistent with this finding, *Mn* exposures induced five histone gene variants *H3C10*, *H2BC13*, *H2AC16*, *H2AC20*, and *H2BC9* of the best ten differentially expressed genes (DEGs). 

The transcriptomics microarray analyses demonstrated that the antioxidant, low molecular weight, neuroprotectant metallothionein-III (MT-3) was among the top *Mn* induced mRNAs in SH-SY5Y cells. MT-3 is a neuroprotectant capable of offsetting AD [69] and PD, as well as biochemically limiting *Mn*-induced damage [58,70]. Like cobalt, zinc, and DFO, the divalent metal *Mn* has been experimentally employed in lung epithelial and adipocytes as a chemical mimetic of hypoxia [52,53]. The data in Figure 6 further validated *Mn* as a hypoxia mimic by the use of RT-PCR, as is consistent with a model for MT-3 providing neuroprotection to SH-SY5Y cells. 

Overall, using the SH-SY5Y model of dopaminergic-transformed neurons, the results from our bioinformatic analysis indicated that *Mn* exposures primarily modulated pathways involved in depurination, HDAC-related neuroprotection, histone modulation, and amyloid fiber formation. Manganese also specifically upregulated specific target genes listed in Table 2, such as the histone genes (as stated above), stress response, transport genes (*MT3*, *AQP10*), and immediate early genes (*ATF3*, *c-Fos*, *EGR1*). Neuronal damage resulting from *Mn* exposure elevated reactive oxygen species (ROS) concentrations, which we reported to be mediated via IRE/IRP control of ferritin (iron storage) and APP (iron efflux by APP/FPN complexes) translational control circuits [7]. Consequentially, ROS surplus by *Mn* can activate NF-κB, which may exacerbate neuronal damage by further elevating concentrations of reactive oxygen species to cause pro-inflammatory reactions [71]. 

As an adaptive cellular response, we note that SIRT1 deacetylases the p65 subunit of NF-κB at its Lys310 residue, an event that could protect cells from apoptosis by reducing transcriptional activity [72]. Overall, the adaptive upregulation of SIRT1 (Table 1) and downregulation of NF-kB pathways is a cellular marker indicating accelerated activation of an ameliorative SIRT1/NF-κB signaling pathway to reduce neuroinflammation [73]. Furthermore, increases in the HDAC expression pathway are specifically involved in this functional pathway, i.e., the Class I HDACs (*HDAC1*, *HDAC3*, *HDAC8*) and *HDAC10*. Notably in neuronal cells, *HDAC1* binds to SIRT1 and provides neuroprotective effects [74]. HDAC expression was increased and involved in neurodegeneration through inflammation [75]. 

Following exposure of manganese to SH-SY5Y cells, there was increased expression of three depurination-related pathways from the eight top *Mn*-induced pathways listed in Table 1. These adaptations are likely in response to damage to adenine and guanine structures. This prompted our report that aspects of *Mn* neurotoxicity could be quelled by urate (Figure 3, Figure 4 and Figure 5). We reasoned that the purine damage can be ameliorated by supplementing urate to repair the transversion mutation; this theory was also supported by our prior publications [45,46]. The purine metabolism pathway we observed after *Mn* perturbations controls the rates of mutation load to affected cells [43,76]. It is noteworthy that increased intracellular urate is the metabolic product of the oral purine supplement inosine and has long been shown to be a neuro-protective antioxidant [42,44,46,47,77]. The *Mn*-induced loss of APP and ferritin we observed here was consistent with our previously reported losses of IRE/IRP-dependent translation of both H-ferritin and APP mRNAs [7]. 

In addition to our own findings, our colleagues have shown that the APP 5′-UTR-directed drugs posiphen and phenserine inhibit APP translation to generate anti-amyloid efficacies [78,79] for the purpose of treating AD as effectively as current anti-inflammatory pathways [80]. Antioxidant and APP-processing pathways also proved to be highly relevant as key therapies for AD [81,82]. However, in younger adults and children, amyloidosis is not a risk factor for developing AD. Therefore, as a model for mitigating metal toxicity and oxidative neurotoxicity in younger age groups, we suggest that antioxidants such as urate might favor APP translation, possibly via the same 5′-UTR, and thus increases of APP/FPN complexes, sufficient to efflux excess embargoed *Mn* and *Fe* in SH-SY5Y cells, as outlined in our previously reported models [7,9]. Here, urate treatments left ferritin expression unchanged, as shown in Figure 3. Thus, the canonical IRE was unresponsive to the intracellular therapeutic presence of urate, while the 5-’UTR specific IRE-type-II of APP mRNA may be tested as being responsive to urate (iron efflux-associated event [9,83]). These events are consistent with the theory that urate could induce neuroprotective APP(s), an event that facilitates FPN-dependent *Fe* efflux and thus counteracted ferroptosis resulting from prolonged cellular exposures to *Mn* [16].

Mn inhibition of APP translation requires further testing to detect decreases in levels of APP(s) as well as c-terminal fragments of APP (APP-CTF-b) which are produced after BACE cleavage. Certainly, APPs and APP-CTF-b fragments of APP have differing consequences to influence cellular iron homeostasis [84,85]. Here, we employed conditions of *Mn* exposure where APP was limited at the translational level so as to limit the appearance of APP(s) and APP CTF [86]. We previously stated that APP 5′UTR translation blockers reduced both fragments in parallel, as detected by 22C11 and A8717 antibodies on Western blots [86]. Thus, *Mn* does not appear to separately induce CTF-b via compensatory activation of BACE in SHSY5Y cells although this remains to be compared in human and mouse cell lines [86]. The biological action of APP CTF fragments might change intracellular iron levels via perturbations to the mitochondrial ATPases that induce lysosomal driven pH changes and inclusion bodies [84,85,87]. Here, lysosomal dysfunction was shown to occur when APP is overexpressed in Down syndrome (triple APP) and in Alzheimer mouse models so as to proportionately also increase APP CTF (see citations [85,88]. By contrast, *Mn* might not promote the appearance of Tyr(682)-phosphorylated APP beta-CTF if shown to induce parallel losses of both of APP and APP CTF. 

Metallothionein-III is a known key neuroprotectant that was noticeably induced in SH-SY5Y cells exposed to manganese. *Mn* is a divalent cation and may well exhibit biology typical of other divalent cations such as cobalt and zinc, which are known hypoxia mimetics [54,66]. Our bioinformatics findings showed that *Mn* induced MT-III mRNA three-fold, while the data in Figure 6 confirmed hypoxic mimetics are active inducers of MT-3 mRNA, which is consistent with published RT-PCR analyses in human adipocytes [53]. In this context, our data in Figure 6 confirm MT-3 mRNAs were 100-fold induced in response to the hypoxia mimetics DFO and zinc.

The bioinformatic findings in this communication therefore provide rationale to reintroduce urate as a neuroprotective agent to treat *Mn* neurotoxicity. In this context, depurination is a chemical reaction of purine deoxyribonucleosides, deoxyadenosine, and deoxyguanosine, in which the beta-n-glycosidic bond is hydrolytically cleaved, releasing the nucleic base, adenine and guanine, respectively [76]. The removal of purine structures elicits a transversion mutation that needs a new purine to repair the DNA [43]. 

Metallothioneins (MTs) are cysteine-rich metal-binding proteins that chelate metals and inhibit oxidative stress, inflammation, and mitochondrial dysfunction induced by metals, while MT-III is reduced in Alzheimer’s disease [69]. Furthermore, MT proteins scavenge free radicals and exhibit anti-inflammatory effects through the suppression of microglial activation; MTs are also critical, as they can act as targets for reducing metal-induced α-synuclein aggregation [58]. Our study showed that MT-3 was increased three-fold following manganese exposure in SH-SY5Y cells. MT-3 is also transcriptionally induced several hundred-fold by our hypoxia mimetics and in [55]. Our current findings in Figure 6 confirm that deferoxamine induced MT-3 mRNA in SH-SY5Y cells by several multiples of 10 while leaving APP mRNA unchanged; also, we determined that there is a putative IRE in the 3′UTR of MT-3 mRNA [89,90]. We therefore conducted validating RT-PCR studies of SH-SY5Y cells dosed with zinc and deferoxamine, agents known to induce hypoxia, and we found these agents, like *Mn*, exhibited similar effects to significantly induce MT-3 mRNA. Overall, like deferoxamine and zinc, manganese is a hypoxia mimetic that significantly upregulates the expression of MT-3 mRNA levels in SH-SY5Y cells.

In the literature, both DFO and manganese can function to induce hypoxia, while DFO is known to upregulate HIF1α in rat brains after injury [91]. In lung cells, manganese induces chemical hypoxia by inhibiting HIF-prolyl hydroxylase. This action interferes with the hydroxylation of HIF-1α, a key post-translational modification for the Von Hippel–Lindau dependent degradation of HIF-1α [52]. Manganese was also shown to mediate the upregulation of HIF-1α protein in Hep2 human laryngeal epithelial cells by activating the MAPK family of enzymes [92]. These manganese-mediated hypoxic conditions appear potentially to play a role in the induction of MT-3, a highly inducible gene in hypoxia [55]. Based on these considerations, Metallothionein-III, like urate, may present new therapeutic angles for treating neurotoxic assaults from manganese overexposures, such as those that occur in children suffering from the effects of industrial pollution or for those suffering from neurodegenerative Manganism [7,26].

At a more global level, *Mn* exposures increase caspase-3 promoter activity through the Sp1-binding regions, increasing the caspase-3 mRNA and triggering the apoptotic chromatin condensation [70], which may be similar to the latent disruption of Sp1 induced by early-life exposure to *Pb* that we have previously reported [3,93,94]. Our results indicate an increase in the expression of core histones; specifically, *H2BC9*, *H2AC20*, *H2AC16*, *H2BC13*, and *H3C10* in SH-SY5Y cells. We note that the increase in histones in response to DNA damage is essential for protecting cells against hydroxyl radical-induced DNA damage [95]. Histone mRNA is the only replication-dependent mRNA in eukaryotes, and is typically expressed during the S-phase of the cell cycle to coincide with DNA replication [96]. It has been shown that cadmium, nickel, and chromium, divalent cations similar to manganese, deplete the stem–loop binding protein (SLBP). This SLBP structure is essential for histone pre-mRNA pre-processing, histone mRNA stabilization and migration, and proper histone expression. This occurs because the histone mRNAs have a stem–loop structure at their 3′ ends, meaning they are not polyadenylated; these stem–loop structures are also highly critical to the histone mRNA regulation. Depletion of the SLBP will increase histone mRNAs outside the S-phase of the cell cycle through elevated polyadenylation. This depletion is also implicated in altered chromatin assembly, supporting the findings that manganese increases chromatin condensation, [97] while compensatory histone expression was observed in Table 2. Furthermore, the polyadenylated genes have been shown to be expressed in response to DNA damage or senescence, which is also present in manganese-induced toxicity, functionally acting as a biomarker of DNA damage [98]. 

Increases occur not only for MT-3 mRNA (Table 2) but also for TfR mRNA levels and these data are included in Table 3 to represent top potential cellular biomarkers for *Mn* toxicity. We re-quantitated our q-RT-PCR data that we had previously reported [7]. In Table 3, TfR mRNA levels are listed to be increased by >7-fold in SH-SY5Y cells when exposed to acute *Mn*Cl_2_ bolus (i.e., 7.6-fold, 10 mM *Mn*Cl_2_ (SD = 0.2219), and 7.8-fold, 100 mM. (SD = 2); each result was obtained after 24 h treatments [7]. Under the same conditions, iron (as ferric ammonium citrate) did not alter TfR mRNA levels in SH-SY5Y cells, while iron chelation with DFO caused a 9.7-fold increase in the steady state of TfR-mRNA levels. We note that TfR protein is readily detected in the serum as a blood-based biomarker and elevated in conditions of anemia. The reference range of serum TfR varies by sex in adults. Normal findings are as follows: Men: 2–5 mg/L, Women: 1.9–4.4 mg/L [99]. Certainly, increases in TfR mRNA expression in certain blood cell types, including lymphocytes, might be tested as a biomarker for *Mn* exposure while, conversely, levels of serum TfR were already reported to be reduced in welding factory workers who were chronically exposed to *Mn* in airborne fumes [67]. RNA stability factors may confound this direct correlation between TfR mRNA and TfR protein levels in the blood of *Mn*-exposed patients [16].

Several immediate early genes (IEGs), *ATF3*, *FOS*, and *EGR1*, were significantly upregulated by manganese exposures. First, *ATF3* encodes the CREB protein family of transcription factors. *ATF3* expression also mediates the mitochondrial stress response [100]. Secondly, *c-Fos* and the associated *c-Jun* potentially form heterodimer AP-1, expressing subsequent survival genes [101]. *c-Fos*, a proto-oncogene, has been determined to show an immediate increase in response to brain injury, and it increases to protect neurons [102]. Thirdly, *EGR1* has been associated with the control of neuronal cell death and inflammation [103,104,105]. Fourth, *AQP10*, a member of the aquaporin family, was significantly upregulated in response to manganese. Fifth, this aquaporin potentially plays a potential role in the transport of glycerol [106]. We provide a highly useful dendrogram that represents the relationships between Mn-induced genes based on their expression patterns (as shown in the heatmap in Figure 2). The dendrogram branches indicate the presence of co-expression. *MT3*, *H2AC16*, and *H2BC13* express similar patterns that potentially function through similar neuroprotective pathways. *FOS*, *EGR1*, and *ATF3* are also biologically related due to the inferred dendrogram relatedness functioning through an immediate early gene pathway. 

Our demonstration of key acute expression effects for *Mn* treatment can be used to interlock with other pathways vital to understanding neurodegeneration. PD is not usually associated with acute toxic responses any more than AD. However, chronic elevation levels of *Mn* have been reported for the movement disturbances of manganism which is distinct from, although often compared with, Parkinson’s disease. Should Mn have a place as an important factor in neurodegenerative disorders, it may be through latent pathways, such as changing epigenetic markers, as has been explored for *Pb* and dementia [27,107,108]. For example, *Mn* overexposure of SH-SY5Y cells alters the methylation of PD-associated genes [109] and induces histone acetylation changes [110]. Thus, we consider this work to be an important foundation in establishing not only acute but potentially long-term effects of *Mn* overexposure in neurological health.

## Figures and Tables

**Figure 1 biomolecules-14-00647-f001:**
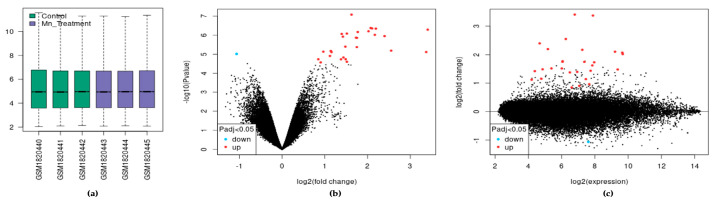
The effect of exposure of SH-SY5Y cells to *Mn*. (**a**) Expression boxplot of the three treatments and three control samples. (**b**) Volcano plot of differential expression and significance by adjusted *p*-value. (**c**) MD plot of differential expression.

**Figure 2 biomolecules-14-00647-f002:**
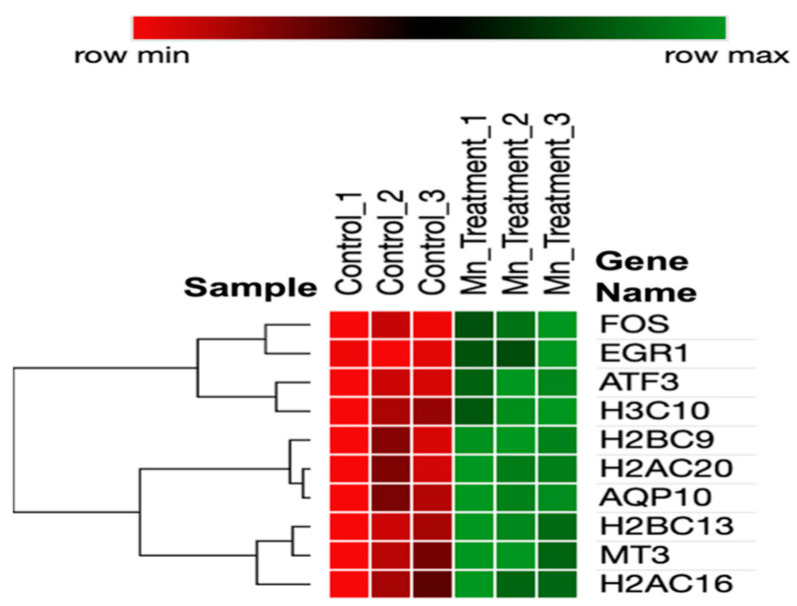
Heatmap of DEGs in chronic manganese exposure.

**Figure 3 biomolecules-14-00647-f003:**
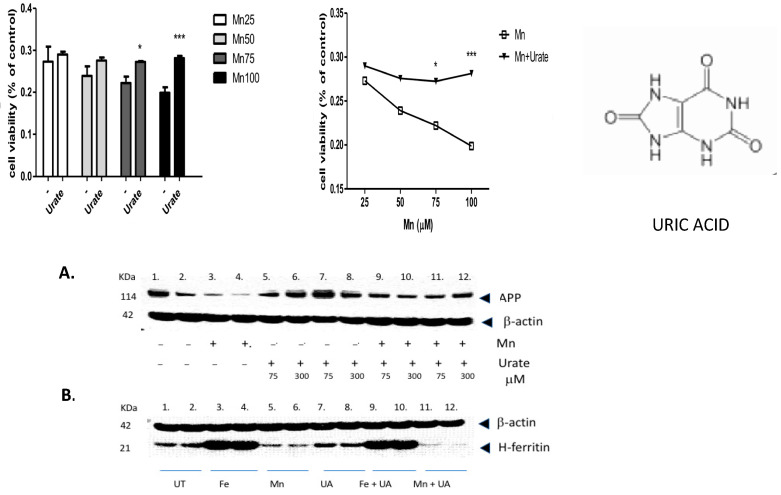
The purine and antioxidant urate offsets manganese-dependent pathways of cell-based toxicity to SH-SY5Y neuroblastoma cells. Top Panels: MTT assay of cell viability: The effects of co-treatment of cells with uric acid (100 mM) on dose-responsive *Mn* toxicity to SH-SY5Y neuroblastoma cells relative to controls (i.e., Plus/minus urate to mitigate toxicity of 25 mM *Mn*Cl_2_, 50 mM *MnCl_2_*, 75 mM *MnCl_2_* and 100 mM *MnCl_2_* to cells for 72 h, N = 6. Bottom Panels: A: Representative Western blot depicting the effects of urate and manganese to modulate APP (FPN-APP complexes export excess iron from cells) in SH-SY5Y neuroblastoma cells. Western blot shows the effect of urate on *Mn*-dependent repression of APP by *Mn* in SH-SY5Y cells (9 × 10^6^ cells) when treated for 48 h, (**A**) (i) Lanes 1–2: control; (ii) Lanes 3–4: 100 µM *Mn* as 100 µM *MnCl_2_*; (iii) Lanes 5–8: 2 sets of increasing UA at 75, 300 µM; and (iv) Lanes 9–12: 2 sets of 100 µM *Mn* and increasing UA at 75, 300 µM. B: Effect of urate towards Fe- and *Mn*-dependent modulation of H-ferritin translation; (**B**) (i) Lanes 1–2: control; (ii) Lanes 3–4: *Fe* as 100µM ferric ammonium citrate (*FAC*); (iii) Lanes 5–6: 100 µM *Mn* as 100 µM *MnCl_2_*; (iv) Lanes 7–8: 100 µM UA; (v) Lanes 9–10: 100 µM *Mn* and 100 µM UA; and (vi) Lanes 11–12: 100 µM *Mn* and 100 µM UA. Full-length Western blot Gels #95 and #97 are provided in the Appendix A section of this paper. * *p* < 0.05 for Student’s *t*-test conducted between *Mn* and *Mn* + UA groups. *** *p* = 0.000165; Student’s *t*-test conducted between *Mn* and *Mn* + UA groups.

**Figure 4 biomolecules-14-00647-f004:**
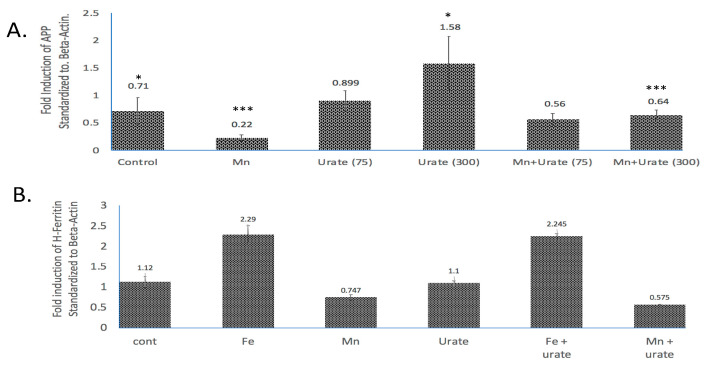
Densitometric quantization of the Western blots shown in Figure 3 after normalization with β-actin. (**A**) Standardized fold modulations (1 = 100%) of the APP/b-actin levels in response to *Mn* dose (100 mM, 48 h) as compared to control and *Mn* treatment in the presence of urate (75 mM and 300 mM urate; see Western blots in Appendix A). Prior to normalization, the data showed β-actin was unchanged by all treatments. (**B**) Standardized fold modulations of the H-ferritin/b-actin levels in response to *Fe* (100 mM FAC, 48 h) and *Mn* (100 mM *MnCl_2_*, 48 h) as compared to controls; *Fe* and *Mn* treatments in the co-presence of 100 mM urate (see Appendix A). Data without normalization with β-actin were unchanged by treatments. To test statistical significance for changes in APP expression, unpaired two-tailed *t*-tests were performed for controls vs. treated lanes and are indicated on the graphs. * & *** *p* ≤ 0.05, i.e., for Cont. vs. UA and for *Mn* vs. *Mn* + UA treatments. Statistical P values reflect the capacity of urate to restore APP expression after its inhibition by Mn treatments, n = 3 independent WB experiments, each in duplicate.

**Figure 5 biomolecules-14-00647-f005:**
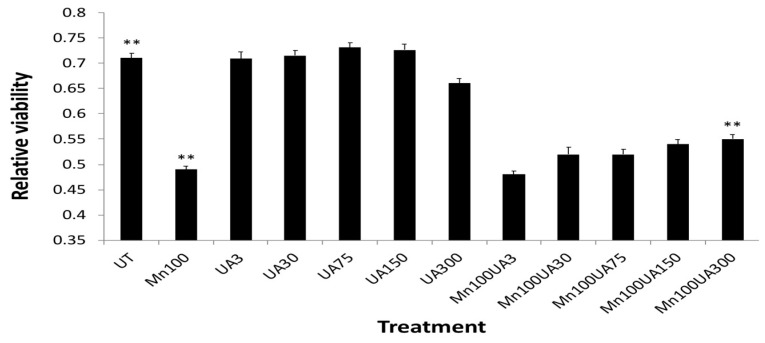
Averaged Urate (@ 3, 30 75, 150 300 mM) pre-treatments afforded a dose graded up to 14%, which increased neuroprotection to SH-SY5Y cells after washout when exposed to *Mn* (100 mM 24 h) (n = 8). ** One-tailed *t*-tests were performed for controls vs. *Mn*-treated, *p* = 4.00 × 10^−8^; *Mn*-treated vs. *Mn* + UA treated, *p* = 3.70 × 10^−2^; control vs. *Mn* + UA treated, *p* = 6.689 × 10^−5^ (N = 8).

**Figure 6 biomolecules-14-00647-f006:**
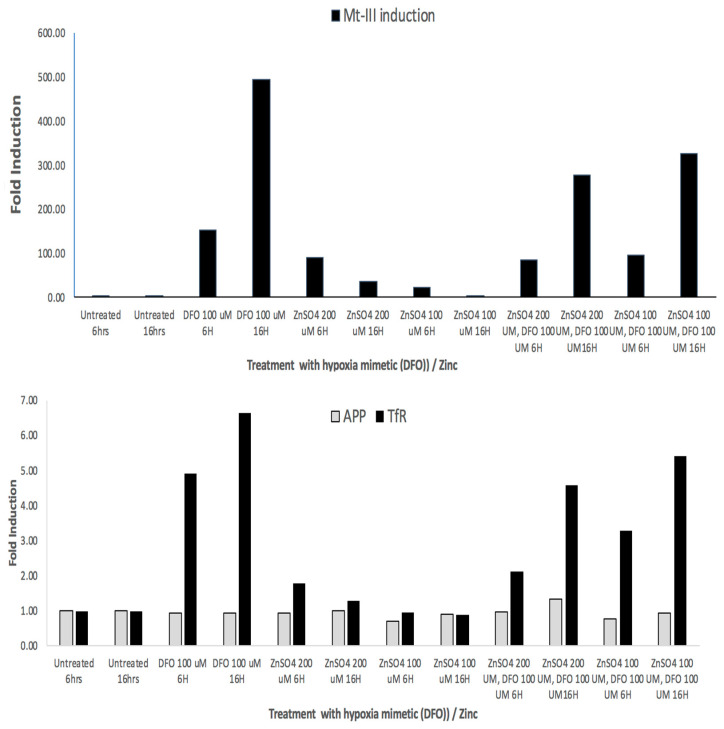
**Top Panel**: RT-PCR: Fold inductions of MT-3 mRNA in response to the hypoxia mimetics DFO and zinc in SH-SY5Y cells. **Second Panel**: RT-PCR showing fold inductions of SH-SY5Y-specific APP mRNA and Transferrin receptor mRNA in response to the hypoxia mimetics *DFO* and *Zn* (negative and positive controls, respectively).

**Table 1 biomolecules-14-00647-t001:** Critical pathways modulated by *Mn* exposure.

Pathway	Regulation	Adj. *p*-Value
SIRT1 negatively regulates rRNA expression	Up	1.51 × 10^−7^
Nucleotide salvage defects	Up	3.96 × 10^−2^
Cleavage of the damaged purine	Up	1.94 × 10^−5^
Depurination	Up	1.47 × 10^−3^
HDACs deacetylate histones	Up	7.82 × 10^−6^
Amyloid fiber formation	Up	9.58 × 10^−5^
Activation of NF-κB as in B cells	Down	4.34 × 10^−2^
Regulated NRF2 gene expression	Down	1.14 × 10^−2^

**Table 2 biomolecules-14-00647-t002:** Top differentially expressed genes (DEGs) modulated by manganese exposure.

Gene Name	Official Gene ID	Fold Change	Adj. *p*-Value
Fos Proto-Oncogene	*c-Fos*	4.5739312	0.006176581
H2B Clustered Histone 9	*H2BC9*	4.20377174	0.00894777
H2A Clustered Histone 20	*H2AC20*	3.36206822	0.006787579
Metallothionein 3	*MT3*	3.35987602	0.01623366
Aquaporin 10	*AQP10*	3.31760548	0.006787579
H2A Clustered Histone 16	*H2AC16*	2.80395203	0.042364419
H2B Clustered Histone 13	*H2BC13*	2.67995046	0.006787579
Activating Transcription Factor 3	*ATF3*	2.62974577	0.00651086
Early Growth Response 1	*EGR1*	2.20370534	0.020852612
H3 Clustered Histone 10	*H3C10*	2.17211627	0.031638477

**Table 3 biomolecules-14-00647-t003:** Potential cellular biomarkers of manganese toxicity in SH-SY5Y cells (100 mM, 24 h).

Gene Name	Fold Changes	Citations
Transferrin Receptor mRNA	Up 7.8-fold, SD = 2	(Venkataramani et al., 2018) Ref. [7]
Serum Ferritin (L-subunit-rich)	Unchanged	(Lu et al., 2005), Ref. [67]
H-Ferritin subunit	Decreased to 5% of control.	(Venkataramani et al., 2018), Ref. [7]
Iron Regulatory Protein 1 (IRP1)	Unchanged	(Venkataramani et al., 2018) Ref. [7]
IRP 1/APP-IRE-Type-II	Up 2-fold, SD = 0.2	To be submitted
Iron Regulatory Protein 2	Decreased to 9.05% of control	(Venkataramani et al., 2018), Ref. [7]
Metallothionein 3	Up 3.36-fold, SD = 0.016	Table 2
Alpha-Synuclein	2-fold decrease	Herein
Alpha-synuclein fibrilization	Increased	(Harischandra et al. 2019), Ref. [68]
Amyloid Precursor Protein	Decreased to 15% of Control 6.7-fold decrease.	Figure 4
Amyloid Precursor Protein mRNA	Up 3.68-fold SD = 0.27	
Translational Inhibition Ratio of APP to APP mRNA	24-fold increase in the index of APP mRNA translation	Cellular Biomarker for *Mn* neurotoxicity.

## Data Availability

The original contributions presented in the study are included in the article/Appendix A, further inquiries can be directed to the corresponding author/s.

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
