# Peer review of "Neuroprotective Strategies and Cell-Based Biomarkers for Manganese-Induced Toxicity in Human Neuroblastoma (SH-SY5Y) Cells"

_biomolecules, 2024, doi:10.3390/biom14060647_

Round 1
Reviewer 1 Report
Comments and Suggestions for Authors
The manuscript is primarily focused on the neurotoxicity caused by Mn and enquires probable measures for neuroprotection utilizing the knowledge from pathway analysis. The authors identified purine metabolism pathways as the central component of Mn-induced neurotoxicity. They introduced Urate as a possible salvage mechanism to mitigate Mn-induced neurotoxicity.
To test their hypothesis, they primarily focused on the increased cell viability and restored levels of APP and H-ferritin following Urate treatment. The experiments and presented data provide just enough evidence to support their claim on the efficacy of Urate. However, since Mn-exposure reports cellular toxicity at multiple axes, the presented data fail to hit the topic on a major note. Experimental evidences providing the levels of additional cellular markers related to Mn-induced toxicity will greatly benefit the candidacy of Urate and increase the interest on the manuscript overall. Therefore, I recommend the manuscript for publication provided that it addresses the cellular levels of additional markers related to Mn-induced toxicity.
Reviewer 2 Report
Comments and Suggestions for Authors
The findings seem interesting; however, the authors need to check both APP and APP-CTF. Additionally, it is strongly suggested to use a neuronal cell line, such as the N2a cell line along with SH-SY5Y.
Reviewer 3 Report
Comments and Suggestions for Authors
Dear Editor/Authors,
The manuscript ID: biomolecules-2833118 titled: “Neuroprotective strategies to mitigate Manganese toxicity to the Human Neuroblastoma (SH-SY5Y) Cell-Based Model” by the authors: Catherine M. Cahill, Sanjan S. Sarang, Rachit Bakshi, Ning Xia, Debomoy K. Lahiri and Jack T. Rogers, present a study aimed to provide a rationale for further testing the ability of urate to mitigate the damage to cell survival caused by manganese. The authors also highlighted the microarray-oriented bioinformatic results showing that metallothionein-III (MT-III) is a prominent Mn-induced transcript and considered the possibility that MT-III could provide further cell-based antioxidant response to attenuate Mn-induced ferroptosis and neurotoxicity. In their experiments, the authors used SH-SY5Y cell lines and 24h Mn treatment. It has been shown that the co-treatment of cells with uric acid mitigated manganese toxicity to SH-SY5Y neuroblastoma cells.
General comments:
Title
The title is appropriate, precise, and clear to readers.
Abstract
Written clearly and understandably. In brief, includes all the elements for understanding MS
Introduction
The authors in the introduction provides basic information on the toxicity of manganese and its role in some neurodegenerative diseases caused by excessive manganese exposure. They focused on urate metabolism as a possible protective mechanism against manganese toxicity.
Materials and methods
The authors presented the bioinformatic models they used in their work. They also described the use of selected cell lines and the methodology of the work. In experiments, adequate and modern methods were used. Lines 12-122 state that the cells are treated identically. It is necessary to describe the experimental design in detail, e.g. duration of treatment, the concentrations used, etc. The subsection Statistical analysis is missing. It is necessary to describe the statistical methods they used to process the data. The Student's t-test is mentioned for the first time in the Results section, line 196, but the statistical methods should be described in the M&M section. It should be described how the p values were adjusted.
Results
The results are clearly written and clearly presented in tables and figures. Hovewer, some explanations are needed.
The authors identified 22 DEGs (Figure 1B) and in Table 2 they presented the top 10 DEGs. Did I understand correctly?
Lines 174-175: significance is needed.
Line 196: The statistical tests used and the meaning of the p-value should be described in M&M.Lines 203-204: For a better understanding of the experimental design, some information, e.g. that the experiment was carried out in duplicate, three separate experiments, etc., should be described in the M&M section.
Discussion
The discussion is based on the results obtained. Authors clearly showed that Mn reduced APP concentration by 3-fold, while the presence of urate increased APP protein concentration by 2-3-fold, so that the presence of co-added urate mitigated Mn inhibition of APP levels to 90 percent of control levels. From eight top Mn-induced pathways, there was an increased expression of three depurination-related pathways after treatment of SH-SY5Y cells.
Supplementary material is adequate.
No specific comments in the text.
Conclusion of the Reviewer
In this MS authors conducted bioinformatic analyses to identify the metabolic pathways modulated by Mn exposures in SH-SY5Ycells which were further experimentaly validated on these cells. Manganese treatment has been shown to modulate key pathways of purine metabolism. One of the key facts is the need to reintroduce urate as a neuroprotective agent to treat Mn neurotoxicity. One of the most important conclusions of this MS is the attempt to decipher the metabolic pathways not only after acute but also after chronic exposure to manganese as it occurs in everyday life.Therefore, I believe that this manuscript opens new insights to answer many previously unresolved questions related to the toxicity not only of manganese, but also of other heavy metals in neurodegenerative diseases.
For all these reasons, I believe that the manuscript ID: biomolecules-2833118 represents a contribution to the general knowledge of the connection of Mn and neurodegenerative diseases.
General conclusion: Acceptable for publication after minor revision.
Round 2
Reviewer 2 Report
Comments and Suggestions for Authors
The authors acknowledge my concern regarding the need for further validation from the N2a cell line, stating, 'This is an excellent suggestion, and such work is planned to be included in a follow-on paper to this Biomolecules (Communication).'
However, they have not yet conducted the experiment.
Comments on the Quality of English LanguageThe authors acknowledge my concern regarding the need for further validation from the N2a cell line, stating, 'This is an excellent suggestion, and such work is planned to be included in a follow-on paper to this Biomolecules (Communication).'
However, they have not yet conducted the experiment.
